# The Incidence of Type 2 Diabetes Mellitus and Weight Gain in People Living with HIV Receiving a Dolutegravir-Based Antiretroviral Therapy in Addis Ababa, Ethiopia: A Pilot Single-Arm Historical Cohort Study

Tariku Shimels [1,*], Arebu Issa Bilal [2], Desta Samuel [1], Desalew Gedamu [1], Eden Solomon [1] and Zewdneh Shewamene [3]

[1] Saint Paul's Hospital Millennium Medical College, Addis Ababa 1271, Ethiopia; samueldesta6@gmail.com (D.S.); desalewgedamu64@gmail.com (D.G.); edensolkassa@gmail.com (E.S.)

[2] Department of Pharmaceutics & Social Pharmacy, School of Pharmacy, College of Health Sciences, Addis Ababa University, Addis Ababa 1176, Ethiopia; issaareb@gmail.com

[3] Population Council-Ethiopia, Addis Ababa 1165, Ethiopia; zeedshow@gmail.com

* Correspondence: tariku.shimels@sphmmc.edu.et; Tel.: +251-912471223

**Abstract:** Introduction: The development of antiretroviral therapy (ART) has immensely improved the quality of life of people living with HIV/AIDS. Despite such a change, concerns continue to persist regarding the safety of the latest drugs added to the regimens. This study aims to evaluate the incidence of type 2 diabetes mellitus (T2DM) and weight gain in individuals receiving antiretroviral therapy containing dolutegravir at a general hospital in Addis Ababa, Ethiopia. Methods: A retrospective cohort study was conducted at RDDMH from 1 February to 30 March 2022. The study included PLHIV who had dolutegravir substituted into their combined regimen in November 2019. Collected data underwent cleaning, entry, and analysis using Statistical Package for Social Sciences (SPSS) v. 26.0 and R programing. Descriptive statistics were employed for univariate and bivariate analysis. The Kaplan–Meier model in R was used to illustrate the hazard function. A significance level of $p < 0.05$ and a 95% confidence interval were employed for statistical reporting. Results: The study followed 185 PLHIV who were on ART who either substituted their previous regimens or initiated a new dolutegravir-based regimen for 12 months. Most were females (59.5%), aged over 38 years (57.5%), married (50.8%), and had lived with HIV for 7 or more years (51.9%). The incidence proportion of T2DM in this sample was 7.0% (95% CI: 3.8–10.3). The age category ($X^2(1, N = 185) = 12.29$, $p < 0.001$) exhibited a statistically significant relationship with the incidence of T2DM. The cumulative rate of T2DM in the age group over 38 years was approximately 15.4%. The pairwise Wilcoxon signed rank test revealed statistically significant differences in BMI scores between time points. Conclusion: This study observed a noteworthy incidence of T2DM among PLHIV receiving a dolutegravir-based first-line ART. Healthcare providers should prioritize early follow-up and management options for PLHIV who are on dolutegravir-based ART regimens.

**Keywords:** ART; dolutegravir; incidence; hyperglycemia; people living with HIV; T2DM





## 1. Introduction

It is apparent that antiretroviral therapy (ART) significantly improves the well-being of individuals living with human immunodeficiency virus (HIV) [1]. Currently, concerns are reported about an increased risk of other chronic conditions, including type 2 diabetes mellitus (T2DM), attributed to both underlying patient factors and certain drugs used in ART regimens [2,3]. In the literature, the incidence of diabetes mellitus (DM) has been linked to factors such as the HIV virus itself [4], co-infection with hepatitis C virus (HCV) [5], and specific classes of ARV drugs like lopinavir, indinavir, stavudine (d4T), didanosine (ddI), and zidovudine (AZT) [6,7].

On the other hand, contradictory findings have been documented, with some studies showing no association between DM incidence and HIV or ART use at all [8,9]. The conflicting results, coupled with variations in study populations and methodological inconsistencies, create confusion regarding any modifiable differences and appropriate preventive measures. Additionally, with the introduction of newer ART regimens, the risk of developing DM may change significantly among PLHIV taking these drugs. Identifying the safest newer agents in a specific context in terms of adverse events becomes crucial.

In Ethiopia, studies on the prevalence of DM among people living with HIV (PLHIV) have been conducted in various settings [10–15]. According to the findings, the cumulative burden of diabetes ranged between 7.1% and 8.8%, underscoring the public health importance of HIV-DM comorbidity in the population. And metabolic syndromes, marked by lipodystrophy, dyslipidemia, and insulin resistance, were reported at 25% [15]. While these studies have provided insights into the overall magnitude and associated factors at a specific point in time, the incidence of DM and the survival time of PLHIV receiving newer ART drugs remain unknown.

The 2018 national consolidated comprehensive guideline for HIV prevention, care, and treatment in Ethiopia recommends the use of tenofovir/lamivudine/dolutegravir (TDF/3TC/DTG) as the preferred first-line regimen for adults [16]. Interestingly, Dolutegravir (DTG), known for its superior effectiveness and safety, has replaced non-nucleoside reverse transcriptase inhibitors (NNRTIs), particularly nevirapine, in many PLHIV [17,18]. Nevertheless, a few studies have raised concerns about potential adverse events, such as hyperglycemia [19,20], and weight gain [21] following the initiation of dolutegravir.

Even though suspected hyperglycemia or confirmed DM is associated with an increased risk of morbidity and mortality in the general population [22,23] and in PLHIV [24,25], determining the incidence and predictive factors among PLHIV on these drugs remains an essential step. The generated evidence can inform current practices, address modifiable factors, and significantly enhance the quality of life and longevity in this population. Considering the possibility of conflicting evidence on the potential contribution of dolutegravir to the progression of T2DM [26], this study aimed to determine the incidence of hyperglycemia and weight gain among PLHIV who were receiving a dolutegravir-based antiretroviral therapy (ART) at a public hospital in Addis Ababa, Ethiopia.

## 2. Materials and Methods

### 2.1. Study Setting, Design, and Period

The research took place at Ras Desta Damtew Memorial Hospital (RDDMH) in Addis Ababa, Ethiopia. RDDMH is a general hospital with 168 beds and 550 staff members, offering inpatient and outpatient services, including antiretroviral therapy (ART) [27]. The study, using a historical cohort design, investigated the T2DM and weight gain incidence among PLHIV on dolutegravir-based ART at RDDMH from 1 February to 30 March 2022.

### 2.2. Population and Eligibility Criteria

The source population comprised all adult individuals (≥14 years old) living with HIV (PLHIV) who had initiated any highly active antiretroviral therapy (HAART) and were receiving care in the current study setting. This demarcation is based on the consideration and linkage of this group to adult ART or other healthcare clinics in the Ethiopian health system. The eligible study population consisted of PLHIV on a dolutegravir-based HAART regimen actively undergoing follow-up at the hospital. These individuals had no history of diabetes mellitus (DM) before initiating dolutegravir (DTG), and their non-nucleoside reverse transcriptase inhibitor (NNRTI)-based regimen had been substituted by dolutegravir from 1st through 30th of November 2019. They were subsequently placed on the same regimen (TDF/3TC/DTG). This represented an open cohort, encompassing eligible PLHIV who started on the new treatment during the specified period.

## 2.3. Sample Size and Sampling Methods

The sample size for this study was determined using the single population proportion formula. As there were no earlier studies in Ethiopia regarding the prevalence of dolutegravir-related hyperglycemia in people living with HIV (PLHIV), the following assumptions were taken into consideration when estimating the minimum sample size: a 14% prevalence of dolutegravir-related hyperglycemia as documented in [28], a 5% type one error, a 95% confidence level (two-tailed test), and a 5% margin of error. As a result, a total of 185 medical records were determined to be necessary for this study. Along with this, a consecutive sampling method was employed to obtain medical records of PLHIV for whom a substitution had been made during the reviewed period.

## 2.4. Variables of the Study

**Outcome variable(s):** T2DM and BMI.
**Independent variables:**
The following factors were evaluated as independent (attributable) variables:
**Sociodemographic characteristics**: these included age at baseline, sex, and occupation.
**Clinical characteristics**: these encompassed a wide array of factors, including World Health Organization (WHO) staging at baseline, CD4 count at baseline, presence of opportunistic infections (OIs) at baseline, baseline weight, baseline blood pressure (BP) level, height, name of the baseline highly active antiretroviral therapy (HAART) regimen, type of current HAART regimen, duration since HIV diagnosis, duration since ART initiation, baseline history of any chronic comorbidity, history of any adverse drug reaction (ADR), history of therapy switch, and history of smoking.

## 2.5. Data Collection Instrument, Procedure, and Quality Assurance

The data collection utilized a structured data extraction format, which was developed based on a review of the relevant literature [29]. Subsequently, the baseline patient characteristics outlined above were extracted. A time-updated version of the covariates was recorded for the year of enrollment, age, CD4+, ART regimen, BMI, and WHO clinical stage at later times of follow-up.

Additionally, BMI was computed as weight in kilograms divided by the square of the height in meters and was further classified into <18.5 (underweight), 18.5–24.9 (optimal), 25–29.9 (overweight), and ≥30 (obese) [30]. The measurements were taken at three time points: at baseline, when the patient was diagnosed with HIV (Time 1); when PLHIV were initiated with a dolutegravir-based regimen (start of this follow-up period) (Time 2); and any time during the follow-up period when the patient experienced the event or was censored (Time 3).

Diagnosis of type 2 diabetes mellitus (T2DM) was considered upon confirmation of any of the following assessments being recorded by an authorized health professional: (1) FPG ≥ 126 mg/dl, as defined in the 2013 American Diabetes Association criteria [31], or (2) initiation of an anti-diabetic medication following a diagnosis of type 2 diabetes mellitus (T2DM). The use of terms like 'hyperglycemia' or 'DM' in this study refers to the measures outlined in this definition. A trained and informed data collector with experience in ART clinical service conducted the data collection. ART registry codes were employed to retrieve medical records on medication and clinical profiles, with the first author overseeing the data collection process on a daily basis. All reports and procedures in this study adhered to the recommendations of the STROBE guidelines (see Supplementary Materials).

## 2.6. Data Analysis

The collected data underwent manual coding, cleaning, and entry into Microsoft Excel. Data analysis was conducted using Statistical Package for Social Sciences (SPSS) v.26.0 and R version 4.2 package. Descriptive statistics were employed to present univariate and bivariate analysis. When the assumptions of binary logistic regression were not met in the analysis of factors related to the hyperglycemia incidence, we employed the Chi-square

test of independence. A one-year follow-up period that spanned from the first visit in the month of November 2019 to October 2020 was considered in this study. The end mark for the follow-up was marked by either the occurrence of an event (the first diagnosis of diabetes mellitus), death due to any cause, loss to follow-up (defined as 6 months past the next visit), or the end of the cohort, whichever occurred first. The hazard function was visualized using the Kaplan–Meier model in R to observe the impact of age on the incidence of the event. The effect of dolutegravir (DTG) on the weight gain of PLHIV was assessed using a repeated measure ANOVA in R. As normality assumptions were not met for all the three time intervals of BMI measurements (Time.1, Time.2 and Time.3), Friedman's non-parametric model was employed. The Wilcoxon signed rank test was used to compare specific BMI pairs. A significance level of $p < 0.05$ and a 95% confidence interval were used for reporting all outputs.

## 3. Results

### 3.1. Characteristics of Participants

A total of 185 charts of PLHIV who either substituted their earlier regimens or initiated a new dolutegravir-based regimen in November 2019 were followed for a period of 12 months. The majority were female (110, 59.5%) and in the age category of over 38 years (107, 57.5%), nearly half were married (94, 50.8%) and had lived with HIV for 7 or more years (96, 51.9%), about two-thirds were normal weight (126, 68.1%), half were either experienced with ART or transferred-in from other health facilities (94, 50.8%), and over one-third (69, 37.3%) were self-employed. Regarding comorbid conditions, the vast majority had no comorbidity (175, 94.6%). Of the 10 cases with comorbidity, hypertension accounted for the most cases (n = 8), followed by deep venous embolism (DVT) (n = 1) and asthma (n = 1). All participants were reported to be in a WHO staging of a healthy state of health (treatment1). And dolutegravir was substituted in those who developed an event (see Table 1).

**Table 1.** Characteristics of study participants who were started on a DTG-based HAART at RDDMH, Addis Ababa, Ethiopia.

| Variable | Category | Frequency | Percent |
|---|---|---|---|
| Sex | | | |
| | Female | 110 | 59.5 |
| | Male | 75 | 40.5 |
| Age category [a] | | | |
| | 38 years or less | 107 | 57.8 |
| | >38 years | 78 | 42.2 |
| Marital status | | | |
| | Single | 52 | 28.1 |
| | Married | 94 | 50.8 |
| | Widowed | 27 | 14.6 |
| | Divorced/separated | 12 | 6.5 |
| Years since diagnosed with HIV | | | |
| | 7 years or less | 96 | 51.9 |
| | >7 years | 89 | 48.1 |
| BMI when patient started DTG | | | |
| | Underweight | 6 | 3.2 |
| | Normal weight | 126 | 68.1 |
| | Overweight | 50 | 27.0 |
| | Obese | 3 | 1.6 |
| ART experience | | | |
| | Naïve | 91 | 49.2 |
| | Experienced/transferred in | 94 | 50.8 |

**Table 1.** *Cont.*

| Variable | Category | Frequency | Percent |
|---|---|---|---|
| Employment status | | | |
| | Self-employed | 69 | 37.3 |
| | Government-employed | 42 | 22.7 |
| | Student | 24 | 13.0 |
| | Unemployed | 18 | 9.7 |
| | Privately employed | 32 | 17.3 |
| Comorbid status | | | |
| | No | 175 | 94.6 |
| | Yes | 10 | 5.4 |

[a] classification was based on the mean of the distribution.

### 3.2. Incidence Proportion and Related Characteristics of T2DM

The incidence proportion of T2DM in the present setting was found to be 7.0% (95% CI: 3.8–10.3). A Chi-square test of independence was performed to identify the presence of relationships between the incidence of an event and patient characteristics, namely, sex, marital status, employment status, comorbid condition, ART experience, age category, years since diagnosed with HIV, and BMI when patient started dolutegravir-based regimen. Accordingly, it was found that only the age category ($X^2$(1, N = 185) = 12.29, $p < 0.001$) showed a statistically significant relationship with the incidence of T2DM in the cohort (Table 2).

**Table 2.** Test of independence between incidence of T2DM and selected characteristics among PLHIV receiving a dolutegravir-based HAART at RDDMH, Addis Ababa, Ethiopia.

| Patient Characteristics | | Incidence of T2DM | | $X^2$ Model Used | $X^2$ Value | df | *p*-Value (Two-Tailed Test) |
|---|---|---|---|---|---|---|---|
| | | No (n = 172) | Yes (n = 13) | | | | |
| Sex | Female | 104 | 6 | Continuity Correction | 0.519 | 1 | 0.471 |
| | Male | 68 | 7 | | | | |
| Employment status | | | | | | | |
| | Self-employed | 64 | 5 | Likelihood Ratio | 8.445 | 4 | 0.077 |
| | Government-employed | 41 | 1 | | | | |
| | Student | 24 | 0 | | | | |
| | Unemployed | 15 | 3 | | | | |
| | Privately employed | 28 | 4 | | | | |
| Comorbid condition | | | | | | | |
| | No | 164 | 11 | Fisher's exact test | NA | NA | 0.149 |
| | Yes | 8 | 2 | | | | |
| Treatment experience | | | | | | | |
| | Naïve | 81 | 10 | Continuity Correction | 3.192 | 1 | 0.074 |
| | Experienced/transferred in | 91 | 3 | | | | |
| Age category | | | | | | | |
| | 38 years or less | 106 | 1 | Continuity Correction | 12.291 | 1 | 0.000 |
| | >38 years | 66 | 12 | | | | |
| Years since diagnosed with HIV | | | | | | | |
| | 7 years or less | 93 | 3 | Continuity Correction | 3.492 | 1 | 0.062 |
| | >7 years | 79 | 10 | | | | |
| BMI when patient started dolutegravir | | | | | | | |
| | Underweight | 5 | 1 | Likelihood Ratio | 1.231 | 3 | 0.746 |
| | Optimal weight | 118 | 8 | | | | |
| | Overweight | 46 | 4 | | | | |
| | Obese | 3 | 0 | | | | |

As marital status initially exhibited a statistically significant association with the incidence of type 2 diabetes mellitus (T2DM), an analysis was conducted to delve deeper into the influence of marital status on the occurrence of T2DM by stratifying it based on age. Given the prevalence of cells with expected counts below 5, the likelihood ratio test was employed for both age groups. The findings indicated no statistically significant association in either the group aged 38 years or younger ($X^2$(3, N = 185) = 1.701, $p = 0.637$) or the group aged over 38 years ($X^2$(3, N = 185) = 9.229, $p = 0.056$) (see Table 3).

**Table 3.** Test of independence between incidence of T2DM and age-stratified marital status among PLHIV receiving a dolutegravir-based HAART at RDDMH, Addis Ababa, Ethiopia.

| Age Category | Marital Status Category | Incidence of T2DM | | X² Model Used | X² Value | df | *p*-Value (Two-Tailed) |
|---|---|---|---|---|---|---|---|
| | | No | Yes | | | | |
| 38 years or less | Single | 45 | 1 | Likelihood Ratio | 1.701 | 3 | 0.637 |
| | Married | 50 | 0 | | | | |
| | Widowed | 6 | 0 | | | | |
| | Divorced/Separated | 5 | 0 | | | | |
| >38 years | Single | 6 | 0 | Likelihood Ratio | 9.229 | 4 | 0.056 |
| | Married | 39 | 5 | | | | |
| | Widowed | 18 | 3 | | | | |
| | Divorced/Separated | 3 | 4 | | | | |

From the descriptive analysis, it was found that most of those at or below the age of 38 were females (75%). And males dominated nearly 62% of the upper age group. However, about 16.7% of the females and 14.6% of the males over the age of 38 were diagnosed with T2DM. The age-stratified incidence of T2DM is plotted in Figure 1 below. It can be observed that until the fourth month following the substitution or initiation of dolutegravir, both groups did not develop T2DM. However, the group in the upper age group was found to more likely to be diagnosed with the disease in the subsequent months, with the cumulative magnitude being 15.4% at the end of the follow-up period (Figure 1).

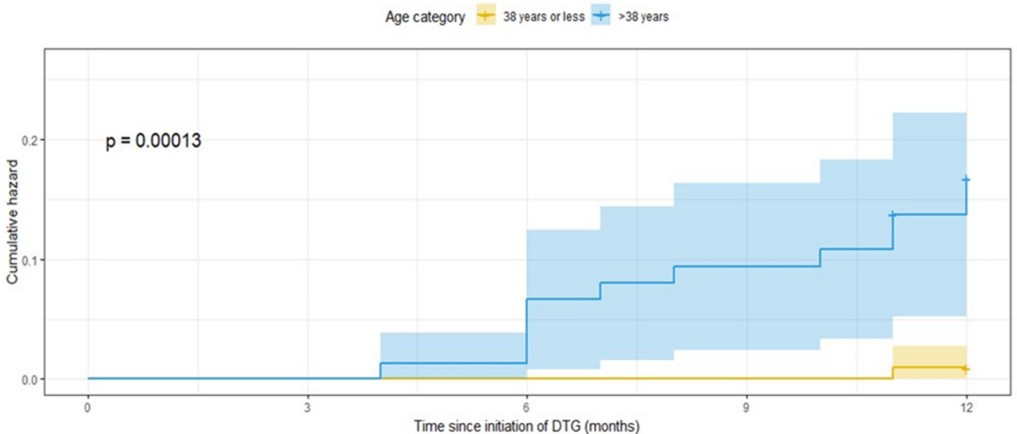

**Figure 1.** Kaplan–Meier hazard plot stratified by age showing T2DM incidence in PLHIV on dolutegravir-based HAART at RDDMH, Addis Ababa, Ethiopia.

### 3.3. Incidence of Weight Gain

A one-way non-parametric repeated measures ANOVA, using Friedman's test in R, was performed. The dependent variable was BMI measured at the three different time points (as a factor), namely, at baseline when the patient was diagnosed with HIV (Time 1), when PLHIV were started on a dolutegravir-based regimen (start of this follow-up period) (Time 2), and any time during the follow-up period when the patient experienced an event or was censored (Time 3). The box plot below shows the distribution of BMI scores (dots on the plot) over the three time intervals (see Figure 2).

A statistically significant difference in BMI scores (dots on the plot) was noted at three different time points (Friedman test, $X^2(2) = 37.49$, $p < 0.001$). Kendall's W was used as the measure of the Friedman test's effect size. Accordingly, the magnitude of the effect size (0.101) was considered small in this evaluation. The pairwise Wilcoxon signed rank test between groups revealed statistically significant differences in BMI scores between Time 1 and Time 3 ($p < 0.001$) and Time 2 and Time 3 ($p < 0.001$). These differences are indicated by asterisks placed between the respective time points on the plot (see Figure 3).

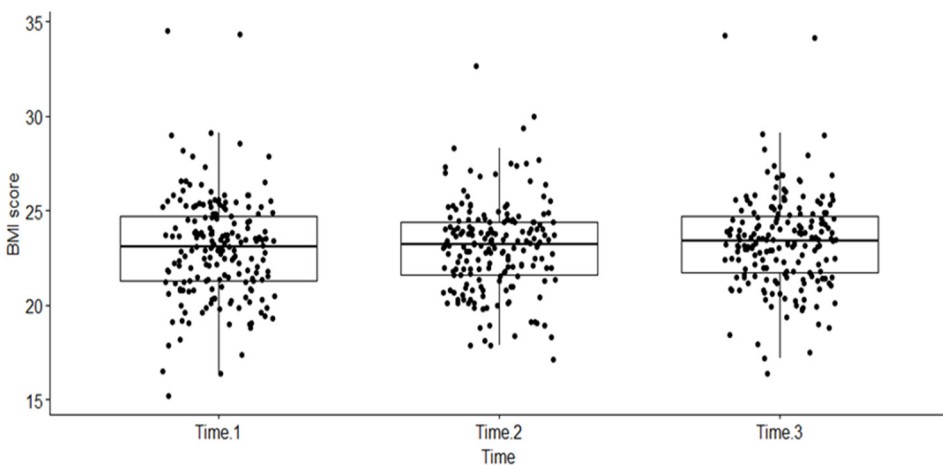

**Figure 2.** Distribution of BMI scores across three time periods among PLHIV treated with dolutegravir-based HAART at RDDMH, Addis Ababa, Ethiopia.

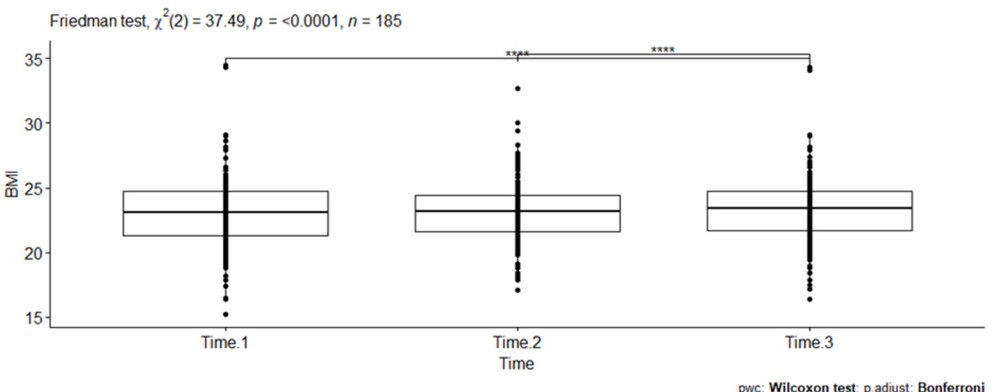

**Figure 3.** Pairwise comparison of Wilcoxon signed rank test between three BMI scores among PLHIV receiving a dolutegravir-based HAART at RDDMH, Addis Ababa, Ethiopia.

## 4. Discussion

The observed incidence of hyperglycemia in the current study setting was approximately 7%, a figure that is consistent with a study on the burden of diabetes in individuals receiving highly active antiretroviral therapy reported from eastern Ethiopia [13]. Despite similarities in the sex and age distribution between the two samples, our study, which had a greater proportion of normal-weight PLHIV when initiating the dolutegravir-based regimen, requires careful consideration due to the measurement type employed. Consequently, the emergence of new cases of type 2 diabetes mellitus (T2DM) within a 12-month follow-up period may signal a potential risk of the drug in altering blood glucose levels in this population. A recent report from Uganda documented dolutegravir-related adverse events, including hyperglycemia, reaching up to 10% [32]. Additionally, only a few cases of diabetes incidence have been reported among PLHIV receiving regimens containing this drug in Ethiopia [19,20]. Evidence from clinical trials has also suggested that integrase-strand transfer inhibitors (INSTIs), including dolutegravir, are generally linked with hyperglycemia. Up to 6% of participants were reported to have experienced a grade 2 event (serum plasma glucose level between 126 and 250 mg/dL) from both dolutegravir and raltegravir [33,34].

The study also explored factors contributing to the incidence of diabetes. However, estimating the potential effect size proved challenging in the logistic regression model due to the limited number of cases with this event. Employing a distribution-free approach via the Chi-square test, a statistically significant correlation between type 2 diabetes mellitus (T2DM) and both marital status and age category was identified. While further investigation

is required to clarify the specific relationship between T2DM and dolutegravir-based highly active antiretroviral therapy (HAART), the association between diabetes risk and marital status appears to be uncommon. To confirm this, an age-stratified analysis examining the relationship between marital status and T2DM incidence revealed no statistically significant association. Conversely, the potential link between age category and diabetes may stem from the fact that nearly all individuals experiencing the event belonged to the age group above the mean. This finding aligns with prior research documenting the heightened association between increasing age and T2DM [35,36].

Assessing the age and sex-wise presentation of the outcome, approximately 16.7% of females and 14.6% of males over the age of 38 were diagnosed with T2DM. The higher proportion in females is attributed to the inverted magnitude of the denominator in the two groups, with females being less likely to be in the upper age group compared to males. Generally, it can be speculated that the risk of diabetes increased in the age group over 38 years, which was noticeably recorded four months following the initiation of a dolutegravir-based regimen and steadily increased over the next months. While aging is a well-studied risk factor for diabetes [37], the result could also be confounded by other factors that alter glucose metabolism in older age, such as changes in body composition and insulin resistance resulting in impaired physiological regulation [38]. In addition, it has been proposed that the insulin resistance in PLHIV receiving INSTIs could be caused by the chelation of magnesium, thereby inhibiting the release and signaling of insulin [39].

The change in BMI of participants was found to be small across the three time points. The mean weight was nearly equal between the first and second measures, with a statistically significant increase noted after the initiation of dolutegravir-based HAART. No stratification was considered in the last BMI (measured after starting dolutegravir), as 93.5% of the cases had completed the follow-up period, and the number of events was small. A five-year retrospective study on the same ART combination reported an average weight gain of 6 kg [40], whereas the study by Bourgi et al. [41] documented the same magnitude of weight gain in 18 months following dolutegravir. The observed slight change in weight gain in the current study might be affected by the short follow-up period of this study and the non-parametric nature of the distribution.

This study is the first of its kind in Ethiopia to provide a contextual understanding of diabetes in people living with HIV (PLHIV) and receiving dolutegravir-based regimens as their first-line highly active antiretroviral therapy (HAART). All appropriate methodological considerations and assumptions have been taken into account to improve the reliability and validity of the findings. Despite thorough efforts and careful considerations, significant limitations may limit the generalizability of our findings. Firstly, no comparison group was considered, making it impossible to compare the effects of different treatments across groups. Secondly, the selection of study participants was based on a one-month inclusion period (PLHIV for whom dolutegravir was started or substituted in the same month and were on the same combined regimen), which might have introduced selection bias, thereby influencing the outcome of interest and between-group analyses. Thirdly, the sample size considered in this study was small, and the effect estimates might not reflect occurrences in other contexts, making the generalization of findings challenging. Lastly, potentially important clinical and virological parameters, such as blood pressure level, random or fasting glucose levels, viral load, and CD4 count, were missing in the majority of the participants and were hence not assessed.

## 5. Conclusions

A noteworthy incidence of type 2 diabetes mellitus was observed among PLHIV receiving a first-line antiretroviral therapy (ART) based on dolutegravir at the current study site. The analysis revealed a statistically significant association between type 2 diabetes mellitus and age category, indicating a progressively higher risk with advancing age. Additionally, a slight increase in the body mass index (BMI) of participants was identified during the 12-month follow-up period following the initiation of dolutegravir.

Healthcare professionals should prioritize timely screening and continuous monitoring of blood glucose markers in this population. It is recommended that prospective studies be undertaken to precisely determine the incidence level, assess the potential role of prognostic factors, such as age, and evaluate the impact on weight gain associated with dolutegravir-based first-line regimens. Furthermore, methodological approaches incorporating proper controls, larger sample sizes, diverse populations, and extended follow-up periods are warranted for a more comprehensive understanding of the implications.

**Supplementary Materials:** The following supporting information can be downloaded at https://www.mdpi.com/article/10.3390/venereology3020008/s1: Table S1: STROBE_checklist.

**Author Contributions:** Conceptualization, T.S.; methodology, T.S.; validation, T.S., D.G., Z.S. and D.S.; formal analysis, T.S.; investigation, E.S. and A.I.B.; data curation, T.S.; writing—original draft preparation, T.S. and A.I.B.; writing—review and editing, T.S., A.I.B., D.G., D.S. and Z.S.; supervision, E.S.; project administration, T.S.; funding acquisition, T.S. All authors have read and agreed to the published version of the manuscript.

**Funding:** This research received no external funding.

**Institutional Review Board Statement:** The study was conducted in accordance with the Declaration of Helsinki and approved by the Institutional Review Board (or Ethics Committee) Saint Paul's Hospital Millennium Medical College (Ref. No: PM22/620).

**Informed Consent Statement:** Informed consent was obtained from all participants involved in the study.

**Data Availability Statement:** All relevant data are contained within the manuscript and its supporting information files.

**Acknowledgments:** We extend our sincere gratitude to all PLHIV and visitors at the ART clinic of RDDMH, whose records were indispensable for the completion of this study. Our appreciation also goes to the ART clinic staff of the hospital, the research department, and the IRB staff of RDDMH and AARHB for their invaluable support in facilitating this particular endeavor.

**Conflicts of Interest:** The authors declare no conflicts of interest.

## Abbreviations

| | |
|---|---|
| ART | Antiretroviral therapy |
| 3TC | Lamivudine |
| AZT | Azathiopurine/Zidovudine |
| BMI | Body mass index |
| CD4+ | Center of Differentiation+ |
| D4T | Stavudine |
| DM | Diabetes mellitus |
| DTG | Dolutegravir |
| FPG | Fasting Plasma Glucose |
| HAART | Highly active antiretroviral therapy |
| HBV | Hepatitis B virus |
| HCV | Hepatitis C virus |
| HIV | Human Immune Virus |
| LPV | Lopinavir |
| LPV/r | Ritonavir-boosted Lopinavir |
| NNRTI | Non-nucleoside reverse transcriptase inhibitor |
| NRTI | Nucleoside reverse transcriptase inhibitor |
| PI | Protease Inhibitor |
| PLHIV | People living with HIV |
| T2DM | Type 2 diabetes mellitus |
| WHO | World Health Organization |

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
