# Peer review of "The Incidence of Type 2 Diabetes Mellitus and Weight Gain in People Living with HIV Receiving a Dolutegravir-Based Antiretroviral Therapy in Addis Ababa, Ethiopia: A Pilot Single-Arm Historical Cohort Study"

_venereology, doi:10.3390/venereology3020008_

Round 1

Reviewer 1 Report

Comments and Suggestions for Authors

There has been lot of concern about the relation between the DTG-based regimen and the weight gain and the increased levels of glicemia.

However, I would be interesting to know how many PLWH had pre-diabetes before the switch and how many of them developed a T2DM, and how many of the ones with multi-morbidities ended with T2DM. I would also recommend to specify the comorbidities (if possible).

Comments on the Quality of English Language

Overall the english could be improved. I would strongly suggest to substituite "client" with "People Living With HIV", "people/person with HIV", "people/ person with AIDS"

Author Response

Dear Esteemed Reviewer,

Thank you very much for your constructive feedback and valuable suggestions.

i) We fully acknowledge your concerns regarding individuals with potential pre-diabetes status. Unfortunately, due to incomplete data, we were unable to assess this outcome/factor comprehensively. We recognize this as one of the limitations of our study.

ii) Regarding the number of individuals with comorbid conditions, we have provided this information, and it's noteworthy that only a small proportion (2 out of 10) developed diabetes mellitus. However, upon analysis using the chi-square test of independence, this association was not found to be statistically significant.

iii) In response to your comment, we have included a list of comorbidities under the subsection 'Characteristics of Study Participants'. It is worth noting that hypertension was the most prevalent comorbidity, observed in 8 out of the 10 cases.

We truly appreciate your insightful input, and we hope that our responses adequately address your concerns.

Best regards, 

Tariku Shimels

Principal Investigator

Reviewer 2 Report

Comments and Suggestions for Authors

The authors have tackled a challenging task and made a sincere effort to do it justice, for which I appreciate their dedication. However, there are several issues with the study, and the results lack novelty.

Major concerns:

1. The absence of a control arm raises concerns about the study design, casting doubt on the reliability of the results. Without a crucial control arm, the study's findings cannot be considered robust.

2. The assertion of a significant increase in T2DM in the Dolutegravir-treated group, particularly with advancing age, prompts the question of whether this rise is solely age-related or linked to the treatment. The methodology for ensuring the distinction needs clarification.

3. The authors claim a statistically significant relationship between marital status and the incidence of T2DM. This association raises skepticism, as it might be correlated with age rather than marital status. Clarification on this matter is essential.

4. The study lacks crucial HIV-related parameters such as viral load, CD4 count, and other immunological measures, as rightly acknowledged by the authors. The absence of these parameters weakens the comprehensiveness and applicability of the findings.

Additional concerns:

1. The authors have inconsistently used various terms such as clients, patients, and People Living with HIV (PLHIV), while the recommended term for this cohort is PLHIV. Standardizing the terminology is advisable for clarity and precision.

2. The criteria for categorizing individuals based on age, specifically choosing 38 years as a threshold, lack explanation. The rationale behind this choice and why other ages were not considered (e.g., 40 or 35 years) should be clarified.

3. The designation of 14 years as the threshold for adulthood, as mentioned in Line 91, requires clarification on how this specific age was determined.

4. The detailed explanation of the hospital from Line 80 to 85 appears to be irrelevant to the study. It is recommended to condense this information for conciseness and relevance to the study.

5. In Line 112, the authors equate hyperglycemia with Type 2 Diabetes Mellitus (T2DM). It's important to note that while hyperglycemia is a significant parameter in T2DM, these terms are not synonymous. The distinction between them needs to be clarified.

6. Repetition is observed in both the clinical characteristics paragraph (Line 117 to 123) and the data collection paragraph (Line 127 to 131). The redundancy should be addressed to enhance clarity and eliminate unnecessary repetition.

7. The content in Lines 152-153 appears unclear and requires rewriting for better comprehension. The authors should revisit and revise this section for coherence.

Comments on the Quality of English Language

The manuscript demonstrates a generally acceptable level of English proficiency, with only minor to moderate editing required in specific sections.

Author Response

Dear Esteemed Reviewer,

Thank you very much for your constructive comments and suggestions.

i) We acknowledge the absence of a control group in our study, which limits its robustness. However, we believe that this single-group cohort provides valuable insights into the uncertainties surrounding the effect of dolutegravir (DTG) on hyperglycemia during the initial months of treatment initiation. We consider this study as a pilot, as there are currently no prior findings in this context. We agree that larger sample sizes and controlled prospective designs are needed to provide further clarification on this topic.

ii) We appreciate your concerns regarding the potential confounding effects of age on the observed risk of diabetes mellitus (DM), particularly in relation to DTG use. While our study lacks a control group, making it challenging to discern the distinct roles of age and DTG, we have highlighted the potential interaction between age and DTG in our discussion. We have also emphasized the need for future studies to include control groups and thoroughly examine the role of age among other factors.

iii) We thank you for raising the issue of confounding by age in our analysis. Upon further examination and controlling for age, we found that marital status has no significant relationship with the incidence of Type 2 diabetes mellitus (T2DM). We chose to report the likelihood ratio test results as it was deemed the most appropriate in situations where the assumptions of the chi-square test were violated. We have included an additional table (Table 3) and made revisions to the results and discussion sections accordingly.

iv) We fully acknowledge the limitation posed by the absence of several biological indicators in our study. Recognizing the importance of HIV-related parameters in understanding the interplay between HIV infection, treatment, and diabetes risk, we have classified this study as a pilot. Moving forward, we are committed to incorporating these crucial parameters into our future research endeavors to enhance the robustness of our findings. We have also recommended that future studies consider these limitations and address them accordingly.

v) We have accepted your suggestion to use the term 'PLHIV' and have avoided redundancy by using terms like 'participants' or appropriate possessive pronouns.

vi) We have clarified that the basis for age classification was the mean of the distribution and have added this information as a footnote to the first table.

vii) We appreciate your clarification regarding the age demarcation of 14 years, and we have added this information to the methods section.

viii) The study setting description has been refined and rephrased as per your suggestion.

ix) We have removed hyperglycemia from the dependent variables list and added a definition for hyperglycemia, clarifying its distinction from Type 2 diabetes mellitus.

x) We have removed duplicates from the data collection section, addressing the redundant use of characteristics.

xi) We have revised the text in the data analysis section as per your comments.

We sincerely appreciate your inputs and hope that we have adequately addressed your concerns.

Thank you once again for your valuable feedback.

Best regards,

Tariku Shimels

Principal Investigator

Reviewer 3 Report

Comments and Suggestions for Authors

The topic of metabolic changes including diabetes and weight gain in response to drugs from the Ingress inhibitor group is a topic of importance and interest in recent years and hence the importance of the current study.

However, there are several problems in this study: the sample is extremely small when the comparison is between 170 patients who did not suffer from diabetes compared to 15...Also, approximately 50% of the study were naive and the DOLUTEGRAVIR treatment was the first treatment of these patients - in this situation it is difficult to attribute metabolic changes in general and especially changes in weight in the first year of treatment which can indicate a "return to normal" in terms of HIV and not necessarily medicinal effects.

In my opinion, a separate analysis should be done for naive or experienced patients.

2 technical notes:

1. In Table 2, I recommend writing the total number of patients for each column (170 and 15) in the headings

2. In the discussion, line 284, it is written that this is the first study that refers to the issue of diabetes in patients with an integrase inhibitor, drawing attention to our recent publication that discusses the same issue-

BMC Infect Dis. 2024 Feb 19;24(1):221. doi: 10.1186/s12879-024-09120-7.

Author Response

Dear Esteemed Reviewer,

Thank you very much for your thoughtful comments and suggestions.

i) If I understand your first concern correctly, you are noting that there was no comparison between groups in this study. Indeed, our study focused on a single cohort, with patients meeting the specified eligibility criteria and without diabetes mellitus at enrollment. We identified a cumulative incidence of 13 cases out of a total of 185, resulting in a proportion of 7%.

ii) It is accurate to highlight that nearly 50% of the patients were treatment-naïve and initiated with a dolutegravir-based regimen. Our results indicated a significant change in BMI between baseline and the last time points. Although this change was not statistically significant, it is noteworthy that the majority of patients in the treatment-naïve group developed diabetes mellitus (DM). Additionally, a vast majority of patients (118 individuals) had a normal weight during treatment with the drug, although this was not statistically significant. As a pilot study, we recognize that these findings may not be conclusive, and we have added a recommendation for future research at the end of the study.

iii) Thank you for bringing Table 2 to our attention. We have revised it accordingly.

iv) I appreciate your comment regarding the reference to a similar article. Our intention in citing the article was to provide context relevant to the situation in Ethiopia. To clarify this, we have added the phrase "in Ethiopia" to ensure the statement is clear.

Thank you once again for your insightful input, and I trust that we have addressed your concerns adequately.

Best regards,

Tariku Shimels

Principal Investigator

Round 2

Reviewer 2 Report

Comments and Suggestions for Authors

I would like to express my appreciation to the authors for addressing most of the comments raised. The manuscript has improved significantly as a result. However, the rationale provided for not including a control group due to the pilot nature of the study is noted. Nonetheless, I maintain the view that the inclusion of an appropriate control group would enhance the robustness of the results and their acceptance within the scientific community.

I commend the authors for their diligence in addressing each comment, particularly for removing the section in which the earlier version of the manuscript suggested a significant role of marital status in the development of DM among PLHIV treated with DTG.

Comments on the Quality of English Language

I find the English language used in the manuscript to be satisfactory.

Author Response

Thank you for your thoughtful comments and acknowledgment of the improvements made to the manuscript. We genuinely appreciate your feedback and are pleased to hear that you recognize our efforts in addressing your concerns.

Regarding the inclusion of a control group, we understand your perspective on enhancing the robustness of the results. While we acknowledge the potential benefits of including a control group, we opted not to include one due to the pilot nature of the study and associated resource constraints. However, we recognize the value of your suggestion and will carefully consider its feasibility for future research endeavors. Following your concern, we've now incorporated the term "single-arm" into the title to clarify the study's design. Additionally, we've made efforts to highlight the limitation of the uncontrolled nature of our study to readers. 

Moreover, as per your suggestion, we have removed the content related to the significant relationship of marital status from the results section. This adjustment ensures that our manuscript accurately presents the findings without overinterpretation.

we trust these revisions can enhance the clarity and transparency of our study, addressing concerns you have raised. We appreciate your guidance and believe these changes will strengthen the manuscript. If you have any further suggestions or feedback, please feel free to share.

Sincerely,

Tariku (Corresponding author)